# Bridging the Chemical Profiles and Biological Effects of *Spathodea campanulat**a* Extracts: A New Contribution on the Road from Natural Treasure to Pharmacy Shelves

**DOI:** 10.3390/molecules27154694

**Published:** 2022-07-22

**Authors:** Łukasz Świątek, Elwira Sieniawska, Kouadio Ibrahime Sinan, Gokhan Zengin, Abdullahi Ibrahim Uba, Kouadio Bene, Magdalena Maciejewska-Turska, Barbara Rajtar, Małgorzata Polz-Dacewicz, Abdurrahman Aktumsek

**Affiliations:** 1Department of Virology with SARS Laboratory, Medical University of Lublin, Chodzki 1, 20-093 Lublin, Poland; barbara.rajtar@umlub.pl (B.R.); malgorzata.polz-dacewicz@umlub.pl (M.P.-D.); 2Department of Natural Products Chemistry, Medical University of Lublin, 20-093 Lublin, Poland; esieniawska@pharmacognosy.org; 3Department of Biology, Science Faculty, Selcuk University, 42130 Konya, Turkey; sinankouadio@gmail.com (K.I.S.); aktumsek@selcuk.edu.tr (A.A.); 4Department of Molecular Biology and Genetics, Faculty of Engineering and Natural Sciences, Kadir Has University, Istanbul 34083, Turkey; abdullahi.iu2@gmail.com; 5Laboratoire de Botanique et Phytothérapie, Unité de Formation et de Recherche Sciences de la Nature, Université Nangui Abrogoua, Abidjan 00225, Côte d’Ivoire; kouadio777@gmail.com; 6Department of Pharmacognosy with Medicinal Plant Garden, Medical University of Lublin, 20-093 Lublin, Poland; magdalena.maciejewska@umlub.pl

**Keywords:** *Spathodea campanulata*, antioxidants, antineoplastic, antiviral, iridoids, bioactive agents

## Abstract

*Spathodea campanulata* is an important medicinal plant with traditional uses in the tropical zone. In the current work, we aimed to determine the chemical profiles and biological effects of extracts (methanolic and infusion (water)) from the leaves and stem bark of *S. campanulata*. The chemical components of the tested extracts were identified using LC-ESI-QTOF-MS. Biological effects were tested in terms of antioxidant (radical scavenging, reducing power, and metal chelating), enzyme inhibitory (cholinesterase, amylase, glucosidase, and tyrosinase), antineoplastic, and antiviral activities. Fifty-seven components were identified in the tested extracts, including iridoids, flavonoids, and phenolic acids as the main constituents. In general, the leaves-MeOH extract was the most active in the antioxidant assays (DPPH, ABTS, CUPRAC, FRAP, metal chelating, and phosphomolybdenum). Antineoplastic effects were tested in normal (VERO cell line) and cancer cell lines (FaDu, HeLa, and RKO). The leaf infusion, as well as the extracts obtained from stem bark, showed antineoplastic activity (CC_50_ 119.03–222.07 µg/mL). Antiviral effects were tested against HHV-1 and CVB3, and the leaf methanolic extract (500 µg/mL) exerted antiviral activity towards HHV-1, inhibiting the viral-induced cytopathic effect and reducing the viral infectious titre by 5.11 log and viral load by 1.45 log. In addition, molecular docking was performed to understand the interactions between selected chemical components and viral targets (HSV-1 DNA polymerase, HSV-1 protease, and HSV-1 thymidine kinase). The results presented suggest that *S. campanulata* may be a bright spot in moving from natural sources to industrial applications, including novel drugs, cosmeceuticals, and nutraceuticals.

## 1. Introduction

Due to their diverse therapeutic properties, herbs have been an integral part of life in many societies worldwide for decades. In many rural African communities, medicinal herbs are the most affordable and accessible treatment method for various illnesses. Approximately 40–45,000 plant species are believed to exist in Africa, of which 5000 are used for medicinal purposes [1]. As an example, *Spathodea campanulata* P. Beauv. is a member of the Bignoniaceae family and is known as the African tulip tree. The plant is widespread in the tropical zone, including West African countries [2], and has great potential for ethnobotanical purposes. For example, its leaves exhibit antidiuretic and anti-inflammatory properties. In addition, its leaves have been reported to act as an anti-inflammatory agent and kidney protectant [2]. In another ethnobotanical report, the plant’s use in the treatment of some diseases, including fever, malaria, diabetes, and pigmentation problems, was stated [3,4]. Due to its ethnobotanical uses, the plant has been the subject of many phytochemical and pharmacological studies. The presence of several bioactive compounds has been reported, including flavonoids, phenolic acids, and iridoids [2,3,5,6,7,8,9]. With regard to its pharmacological properties, a wide range of biological properties, from antioxidant to antiviral, has been declared [2,3,5,10,11,12,13,14,15]. Interestingly, the aqueous leaf extract of *S. campanulata* was found to exert potent larvicidal and pupicidal properties and induce morphological deformities in *Aedes aegypti*, a vector of dangerous pathogenic viruses, including chikungunya, dengue, Mayaro, yellow fever, and Zika viruses [16]. The active fractions and sub-fractions of *S. campanulata* stem bark extract were shown to have activity against *Helicobacter pylori* [5]. *S. campanulata* was also found to exert anti-inflammatory properties [17] and hypoglycaemic and anti-HIV activities [18]. The biological properties of *S. campanulata* have also been tested *in vivo*. Leaf methanolic extract was reported not to induce acute toxicity when administered orally to mice (500–4000 mg/kg). Methanolic extract from *S. campanulata* leaves was also shown to have rapid, long-lasting, and significant sedative and anxiolytic activities in mice [19]. In another study, ethyl acetate and methanolic extracts from *S. campanulata* flowers (200 and 400 mg/kg) showed an antidepressant potential in two behavioural despair tests in mice, and their primary active iridoid ingredients—spatheoside A and spatheoside B—were found to inhibit the monoamine oxidase A (MAO-A) enzyme in a molecular docking analysis [3]. *S. campanulata* ethyl acetate extract (100, 200, and 400 mg/kg) was also found to provide chemopreventive activity by improving the morphological characteristics of sperm as well as antioxidant status in lead acetate-induced testicular toxicity in male Wistar rats [20]. Other in vivo studies have revealed healing activity in burn wounds [21] and excision and infected excision (*Staphylococcus aureus*) wounds [22] and anticonvulsant activity [23].

In view of the above, in the presented research, we aimed to combine the chemical profiles and biological effects of the extracts (methanolic and aqueous (infusion)) from the leaves and stem bark of *Spathodea campanulata*. The chemical profiles of the tested extracts were characterized using the LC-ESI-QTOF-MS method. The biological properties dataset was obtained from antioxidant, anti-enzymatic, antineoplastic, and antiviral assays. The evaluated antioxidant properties included radical scavenging, reducing power, and metal chelating. Amylase, glucosidase, tyrosinase, and cholinesterase were selected for the anti-enzymatic assays. The antineoplastic activity of the extracts was tested on three cancer cell lines of different origins (pharyngeal, cervical, and colon cancers), and green monkey kidney cells were used to represent normal cells. In addition, antiviral activity towards human herpesvirus type 1 (HHV-1) and coxsackievirus B3 (CVB3) replication in VERO cells was assessed. The knowledge obtained here can open new avenues for multifunctional applications with the use of *S. campanulata.*

## 2. Results and Discussion

### 2.1. Chemical Characterization of the Tested Extracts

The total phenolic and flavonoid contents of *S. campanulata* extracts determined using spectrophotometric assays are given in Table 1. The highest total phenolic content was detected in the methanol extract from leaves, with 89.39 mg GAE/g, followed by the leaves-infusion (50.31 mg GAE/g), stem bark-infusion (31.72 mg GAE/g), and stem bark-methanol (23.58 mg GAE/g) extracts. However, the order for total flavonoid content was: leaves-infusion (26.96 mg RE/g) > leaves-methanol (6.32 mg RE/g) > stem bark-infusion (3.70 mg RE/g) > stem bark-methanol (2.38 mg RE/g). Apparently, the leaf extracts contained more total bioactive compounds than the stem bark extracts. Similarly, in a recent study by Santos, Minatel, Lima, Silva, and Chen [13], the leaf extracts of *S. campunalata* collected from Brazil had higher concentrations of total phenolics than flower and nectar extracts. In another study [24], total phenolic content was higher in the ethanol extract of *S. campanulata* leaves compared to chloroform and ethyl acetate extracts. Similar results were also reported by Diane et al. [25], who found the highest levels of total phenolic and flavonoid contents in the ethanol extract of *S. campanulata* extracts. Although colorimetric assays provide simple, inexpensive, and rapid results, their authenticity is declining by the day. One of the main problems with these assays is that certain components do not react with the reagents used in the assays [26]. Thus, the results obtained can become suspect and require further chromatographic techniques to confirm. To confirm our results, we analysed the extracts using the LC-ESI-QTOF-MS method.

The qualitative analysis in negative mode revealed the presence of 57 constituents, which are listed in Table 2 following their elution order. UV-VIS detection was used to distinguish each class of bioactive compounds identified in *Spathodea* extracts (see Figure 1). In the second step, an in-depth identification was performed based on accurate mass measurements and the interpretation of characteristic molecular and product ions acquired using the Agilent MassHunter software (Table 2). Their retention times (Rt), UV spectra, and fragmentation behaviour were compared with publicly available data from the literature (Appendix A). Flavonoids and phenolic acids, including iridoids bearing the caffeic acid moiety, were the prevailing class of biologically active polyphenols identified in the methanolic leaf extract. Although the compounds identified in the methanolic and aqueous extracts of leaves were almost the same, the intensity of the detected signals was significantly higher in the case of the methanolic extract. Moreover, hydroxybenzoic acid, methylgallate, spatheoside C, isoquercetin, ferulic acid, quercetin-*O*-dihexoside, di-*O*-caffeoylcatalpol isomers, quercetin, and spathodic acid were not detected in the infusions, indicating that the activity of the methanolic extract could also be related to the presence of these molecules in the sample. As can be seen in Figure 2a, a smaller number of secondary metabolites was identified in the stem bark extracts. The identified metabolites were phenolics, showing that these kinds of compounds were less abundant in stem bark, and underlining the fact that the most active sample was the richest in flavonoids, phenolic acids, and iridoids. The presented results indicate that the phytochemical profile, and thus biological activity, strongly depend on the part of the plant and extraction solvent used.

Out of the six phenolic acids detected in the *S. campanulata* extracts, only two were identified as hydroxybenzoic acids (**5** and **8**). According to the signals recorded on the BPC, caffeic acid (**12**) yielded the highest peak intensity in all examined samples. Two caffeic acid derivatives, namely caffeoyl-glucopyranoside (**9**) and caffeoyl dihexoside (**37**), were identified in the leaf methanolic and infusion extracts due to the observed neutral loss of the hexose moiety, resulting from the cleavage of the *O*-glycosidic bond [27]. Among other hydroxycinnamic acids, small amounts of ferulic acid (*m*/*z* 193.0490) (**34**) were detected in the leaf and stem bark methanolic extracts in 25 min. This compound was characterized by the diagnostic product ions at *m*/*z* 178.0242, 161.0214, 149.0535, and 134.0355 [28]. Based on the literature data regarding *Spathodea* phytochemicals, many of the detected constituents were indeed iridoids [29]. Ajugol (**6**), for instance, was identified in all samples, while loganic acid (**7**) was found only in leaf extracts, similar to their corresponding derivatives, 6′-*O*-6-*O*-(E)-caffeoylajugol (**49**) and *trans*-caffeoyl-loganic acid (**22**), respectively, which were identified exclusively in the leaf methanolic and infusion extracts [30,31]. On the other hand, catalpol, reported previously by Gouda et al., was absent in all studied extracts (Gouda 2009). However, its derivatives (compounds **14**, **23**, **46**, and **48**) were tentatively identified. For two of these (**14** and **23**), the precursor [M−H]^−^ ions at *m*/*z* 523.14 were in accordance with an empirical molecular formula of C_24_H_28_O_13_; however, their fragmentation pattern differed significantly. Intensive fragments at *m*/*z* 179.03, 161.02, and 135.04 observed in the MS/MS spectra of both compounds corresponded to caffeic acid [27]. However, for compound **23**, which was assigned as 6-*O*-caffeoylcatalpol (verminoside), the product ion at *m*/*z* 361.0851 (for catalpol) supplied evidence for the substitution of caffeoyl at the C-6 position (Li et al. 2014). For compound **14**, which was eluted earlier, the presence of the fragment ion at *m*/*z* 323.0679, resulting from the loss of the iridoid aglycon (C_9_H_12_O_5_) followed by the cross-ring cleavage of glucose (ions at *m*/*z* 281.0622 and 221.0368), enabled its identification as 6′-*O*-caffeoylcatalpol, previously reported in *Neopicrorhiza scrphulariiflora* roots [32]. Another two compounds, **46** and **48**, were found exclusively in the leaf methanolic extract. The deprotonated molecular ion at *m*/*z* 685.17 and a molecular formula of C_33_H_34_O_16_ showed an increase of 162 Da compared to verminoside, suggesting the presence of one additional caffeoyl or glucosyl moiety in their structures. The similarity of the fragmentation to that observed for compounds **14** and **23** allowed their tentative identification as isomers of di-*O*-caffeoylcatalpol [33]. The presence of spatheoside A (**13**) and spatheoside C (**17**), isolated previously from *Spathodea campanulata* P. Beauvais leaves [30,34], was confirmed in all studied samples. Derivatives of spatheoside A (**15** and **38**) were tentatively identified in 20.484 and 27.16 min, respectively. In terms of spatheoside B (6-*O-trans*-caffeoyl-asystasioside E), the results of the LC/MS study revealed a set of four peaks—**20**, **24**, **30**, and **33**—with the same molecular formula of C_24_H_29_ClO_13_. All the compounds showed an abundant deprotonated molecular [M−H]^−^ ion at *m*/*z* 559.12, which followed a similar pattern with further fragmentation (see Table 2). Therefore, due to the insufficient data obtained, compounds **20**, **24**, **30**, and **33** were tentatively identified as isomers of spatheoside B.

With regard to flavonols, a number of quercetin and kaempferol-*O*-glycosides were identified predominantly in the leaves of *S. campanulata*. The type of the sugar unit (hexoside or pentoside) was deduced from the difference in the mass of the deprotonated molecular ion and the masses of the corresponding product ions detected in the MS/MS spectra [35]. In addition to quercetin (**50**), isoquercetin (**26**), and quercetin-*O*-dihexoside (**41**), found only in the methanolic leaf extract, a number of quercetin conjugates were characterized in both leaf extracts [36]. Their chemical structures were tentatively proposed as quercetin-3-*O*-apiosylrutinoside (**18**), rutin (**25**), quercetin-3-*O*-pentosyl-pentoside (**27**), quercetin-*O*-(pentoside-hexoside)-*O*-hexoside (**36**), quercetin-*O*-arabinoside-glucoside-*O*-rhamnoside (**42**), and quercetin-*O*-arabinoside-glucoside-*O*-glucuronide (**43**) based on their retention time, the dominant ion at *m*/*z* 301 and/or 300, the neutral loss of small units, and a *retro*-Diels–Alder (*r*DA) ring-opening mechanism, resulting in the diagnostic ion at *m*/*z* 151 (^1.3^A^−^) [36,37]. Quercetin-3-*O*-(2-*O*-β-d-xylopyranosyl)-β-D-galactopyranoside (**21**) was the only additional compound detected in the stem bark extracts. The dominant product ions at *m*/*z* 285 and/or 284 observed in the MS/MS spectra of compounds **28**, **31**, **32**, **35**, **39**, **44**, and **45** suggested the same aglycone structure, resulting from the neutral loss of sugar moieties (-146Da, -162 Da, -176 Da, and -132 Da) from deprotonated molecular ions. The strongly informative product ion at *m*/*z* 151 confirmed the dihydroxyl substitution in the A-ring of the aforementioned compounds, whereas the characteristic ions at *m*/*z* 179, 161, and 135 detected in the MS/MS spectra of compounds **39, 44,** and **45** confirmed the existence of a caffeoyl moiety in their structure [27,36]. Taking the above into account, compounds **28**, **31**, **32**, **35**, **39**, **44**, and **45** were proposed to be kaempferol 3-*O*-(2-*O*-β-d-xylopyranosyl)-β-d-galactopyranoside, kaempferol-*O*-rutinoside, kaempferol-*O*-sophoroside-*O*-glucoside, kaempferol-*O*-glucuronide, kaempferol-*O*-caffeoyl-pentoside-*O*-hexoside, kaempferol-*O*-caffeoylglucoside, kaempferol-*O*-(caffeoylglucoside)-*O*-rhamnoside, and kaempferol-*O*-(pentoside-hexoside)-*O*-deoxyhexoside, respectively [36,37].

In terms of flavones, four compounds were identified in the leaf methanolic and infusion extracts. By comparing their retention time and fragmentation behaviour with the available literature data, compound **54** was assigned as apigenin; **51** as luteolin; compound **29**, eluted in 23.924 min, as luteolin-*O*-hexoside; and compound **55** as tetrahydroxyflavone, due to insufficient data [38,39,40].

One triterpene (**56**) appeared in about 43.761 min in the chromatograms recorded for almost all extracts except the leaf infusion. Its deprotonated [M−H]^−^ ion at *m*/*z* 487.3429 was consistent with the empirical molecular formula of C_30_H_48_O_5_ and might correspond to spathodic acid, which was described previously only in *S. campanulata* stem bark [5,41].

Organic acids, including malic (**2**), citric (**3**), and quinic acid (**4**), were primarily found in leaves [42]. However, low intensities of compounds **3** and **4** were also found in the stem bark infusion. Lastly, two fatty acids, trihydroxy-octadecadienoic acid (**52**) and trihydroxy-octadecenoic acid (**53**), were identified in all studied extracts except for the stem bark infusion, which did not contain hydroxy octadecatrienoic acid (**57**).

**Table 2 molecules-27-04694-t002:** Spectral characteristics of compounds present in the studied samples.

Comp. No.	Tentative Identification	Rt (min)	Molecular Formula	[M−H]^−^ (*m*/*z*)	Product Ions (*m*/*z*)	Extracts	References
1.	Sucrose	1.569	C_12_H_22_O_11_	341.1035	179.0530; 161.0270; 135.0455; 119.0314; 89.0226	Leaves-MeOH Leaves-infusion	[43]
2.	Malic acid	2.049	C_4_H_6_O_5_	133.0110	115.0040; 89.0235; 71.0135	Leaves-MeOH Leaves-infusion	[44]
3.	Citric acid	2.409	C_6_H_8_O_7_	191.0187	173.0064; 154.9955; 111.0084; 87.0090	Leaves-MeOH Leaves-infusion Stem bark-infusion	[42]
4.	Quinic acid	4.206	C_7_H_12_O_6_	191.0540	173.0425; 101.0591; 85.0649	Leaves-MeOH Leaves-infusion Stem bark-infusion	[44]
5.	Dihydroxybenzoic acid	7.59	C_7_H_6_O_4_	153.0143	109.0302; 108.0225; 91.0172	Leaves-MeOH Leaves-infusion Stem bark-MeOH Stem bark-infusion	[44]
6.	Ajugol	7.802	C_15_H_24_O_9_	347.1358	303.1342; 123.0783; 185.0829; 167.0704	Leaves-MeOH Leaves-infusion Stem bark-MeOH Stem bark-infusion	[30]
7.	Loganic acid	9.121	C_16_H_24_O_10_	375.1251	213.0758; 194.8742; 169.0859; 151.0761; 125.0603	Leaves-MeOH Leaves-infusion	[30]
8.	Hydroxybenzoic acid	9.840	C_7_H_6_O_3_	137.0228	108.0218; 109.0287	Leaves-MeOH Stem bark-MeOH	[44]
9.	Caffeoyl-glucopyranoside	10.080	C_15_H_18_O_9_	341.0877	281.0675; 179.0364; 161.0247; 133.0282; 135.0445	Leaves-MeOH Leaves-infusion	[31]
10.	Methylgallate	11.110	C_8_H_8_O_5_	183.0279	168.0064; 124.0160; 78.0117	Leaves-MeOH	[45]
11.	Iridoid compound	13.721	-	459.1586	281.0622; 279.1173; 179.0330; 135.0459	Leaves-MeOH Leaves-infusion	
12.	Caffeic acid	14.694	C_9_H_8_O_4_	179.0323	135.0424	Leaves-MeOH Leaves-infusion Stem bark-MeOH Stem bark-infusion	[45]
13.	6-*O-trans*-caffeoyldecinnamoyl Globularimin (spatheoside A)	18.698	C_24_H_30_O_14_	541.1554	179.0352; 161.0242; 135.0451	Leaves-MeOH Leaves-infusion Stem bark-MeOH Stem bark-infusion	[30]
14.	6′-*O*-Caffeoylcatalpol	19.189	C_24_H_28_O_13_	523.1473	323.0679; 281.0643; 221.0368; 179.0359; 161.0301; 135.0421	Leaves-MeOH Leaves-infusion	[32]
15.	Derivative of spatheoside A	20.484	-	657.1810	541.1523; 179.0358; 135.0459	Leaves-MeOH Leaves-infusion	
16.	Quercetin diglucoside	20.927	C_27_H_30_O_17_	625.1344	301.0269; 300.0225; 271.0181; 255.0193; 178.9903; 151.0008	Leaves-MeOH Leaves-infusion	[44]
17.	6-*O-trans*-caffeoyl-5,7-bisdeoxycynanchoside (spatheoside C)	21.347	C_24_H_30_O_13_	525.1595	345.0935; 179.0313; 161.0217; 135.0423	Leaves-MeOH Stem bark-MeOH Stem bark-infusion	[30]
18.	Quercetin-3-*O*-apiosylrutinoside	21.551	C_32_H_38_O_20_	741.1808	609.1364; 591.1358; 475.0774; 343.0367; 300.0235; 271.0228; 178.9960; 150.9969	Leaves-MeOH Leaves-infusion	[44]
19.	Iridoid compound 2	21.651	-	535.1568	491.1581; 341.0895; 323.0786; 179.0363; 161.0256; 135.0471; 133.0299	Leaves-MeOH Leaves-infusion	
20.	6-*O-trans*-caffeoyl-asystasioside E (spatheoside B) isomer 1	21.730	C_24_H_29_ClO_13_	559.1241	523.1395; 361.0853; 179.0300; 161.0202; 135.0401	Leaves-MeOH Leaves-infusion Stem bark-MeOH Stem bark-infusion	[30]
21.	Quercetin-3-*O*-(2-*O*-β-d-xylopyranosyl)-β-d-galactopyranoside	22.186	C_26_H_28_O_16_	595.1271	523.1400; 445.0642; 300.0233; 271.0206; 255.0262; 178.9963; 151.0000	Leaves-MeOH Leaves-infusion Stem bark-MeOH Stem bark-infusion	[13,30]
22.	6ʹ-*O*-trans-caffeoyl-loganic acid	22.785	C_25_H_30_O_13_	537.1563	323.0662; 179.0309; 161.0247	Leaves-MeOH Leaves-infusion	[30]
23.	6-*O*-caffeoylcatalpol (verminoside)	22.845	C_24_H_28_O_13_	523.1431	361.0851; 343.0808; 179.0337;161.0241; 135.0458;133.0311	Leaves-MeOH Leaves-infusion Stem bark-MeOH Stem bark-infusion	[30,31]
24.	6-*O-trans*-caffeoyl-asystasioside E (spatheoside B) isomer 2	22.941	C_24_H_29_ClO_13_	559.1268	523.1422; 361.0940; 179.0320; 161.0229; 135.0443; 133.0367	Leaves-MeOH Leaves-infusion Stem bark-MeOH Stem bark-infusion	[3]
25.	Rutin	23.324	C_27_H_30_O_16_	609.1439	300.0225; 271.0202; 151.0040	Leaves-MeOH Leaves-infusion	[38,46]
26.	Isoquercetin	23.742	C_21_H_20_O_12_	463.0861	300.0247; 271.0164; 255.0283; 151.0001	Leaves-MeOH	[38,46]
27.	Quercetin-3-*O*-pentosyl-pentoside	23.744	C_25_H_25_O_15_	565.1154	300.0223; 271.0215; 178.9841; 151.0006; 116.9266	Leaves-MeOH Leaves-infusion	[44]
28.	Kaempferol 3-*O*-(2-*O*-β-d-xylopyranosyl)-β-d-galactopyranoside	23.984	C_26_H_28_O_15_	579.1306	285.0349; 284.0282; 255.0252; 178.9957; 151.0011	Leaves-MeOH Leaves-infusion Stem bark-MeOH Stem bark-infusion	[30]
29.	Luteolin-*O*-hexoside	23.924	C_21_H_20_O_11_	447.0888	285.0359; 284.0282; 151.0022; 133.0258	Leaves-MeOH Leaves-infusion	[36]
30.	6-*O-trans*-caffeoyl-asystasioside E (spatheoside B) isomer 3	24.163	C_24_H_29_ClO_13_	559.1226	523.1405; 179.0313; 161.0212; 135.0421	Leaves-MeOH Leaves-infusion Stem bark-MeOH Stem bark-infusion	[30]
31.	Kaempferol-*O*-rutinoside	24.463	C_27_H_30_O_15_	593.1475	284.0274; 255.0288; 150.9984	Leaves-MeOH Leaves-infusion	[46]
32.	Kaempferol-*O*-sophoroside-*O*-glucoside	24.643	C_33_H_40_O_21_	771.1692	609.1393; 285.0369; 255.0227; 150.9955;	Leaves-MeOH Leaves-infusion	[44]
33.	6-*O-trans*-caffeoyl-asystasioside E (spatheoside B) isomer 4	24.163	C_24_H_29_ClO_13_	559.1214	523.1433; 361.0930; 179.0342; 161.0238; 135.0453	Leaves-MeOH Leaves-infusion Stem bark-MeOH Stem bark-infusion	[30]
34.	Ferulic acid	25.003	C_10_H_10_O_4_	193.0473	178.0242; 161.0214; 149.0535; 134.0355	Leaves-MeOH Stem bark-MeOH	[28]
35.	Kaempferol-O-glucuronide	25.123	C_21_H_18_O_12_	461.0702	285.0360; 151.0008	Leaves-MeOH Leaves-infusion	[44]
36.	Quercetin-*O*-(pentoside-hexoside)-*O*- hexoside	26.225	C_32_H_38_O_21_	757.1526	595.1228; 301.0280; 300.0236; 179.9978; 161.0208; 151.0013	Leaves-MeOH Leaves-infusion	
37.	Caffeoyl dihexoside	26.333	C_21_H_28_O_14_	503.1141	341.0839; 281.0627; 251.0531; 21.0442; 179.0324; 161.0220; 135.0414	Leaves-MeOH Leaves-infusion	[47]
38.	Caffeoyl/glucosyl spatheoside A	27.160	C_30_H_40_O_19_	703.1859	541.1500; 179.0317; 161.0219; 135.0415	Leaves-MeOH Leaves-infusion	
39.	Kaempferol-*O*-caffeoyl-pentoside-*O*-hexoside	27.190	C_35_H_34_0_18_	741.1812	579.1459; 455.1247; 285.0438; 184.0367; 179.0368; 161.0271; 151.0063; 135.0468	Leaves-MeOH Leaves-infusion	
40.	Trihydroxyflanon-*O*-glucuronide	27.400	C_21_H_18_O_11_	445.0733	269.0406	Leaves-MeOH Leaves-infusion	[44]
41.	Quercetin-*O*-dihexoside	27.520	C_30_H_26_O_15_	625.1165	463.0899; 301.0303; 300.0196; 271.0178; 178.9977; 150.9979	Leaves-MeOH	[38]
42.	Quercetin-*O*- arabinoside-glucoside-*O*-rhamnoside	28.110	C_32_H_38_O_20_	741.1812	595.1311; 301.0324; 300.0280; 178.9983; 151.0041	Leaves-MeOH Leaves-infusion	
43.	Quercetin-*O*-arabinoside-glucoside-*O*-glucuronide	28.119	C_32_H_36_O_22_	771.1725	595.1144; 300.0237; 271.0192; 255.0354; 178.9916; 150.9941	Leaves-MeOH Leaves-infusion	
44.	Kaempferol-*O*-caffeoylglucoside	28.444	C_30_H_26_O_14_	609.1353	447.0937; 285.0407; 179.0356; 161.0241; 151.0049; 135.0452	Leaves-MeOH Leaves-infusion	[44]
45.	Kaempferol-*O*-(caffeoylglucoside)-*O*-rhamnoside	28.503	C_36_H_36_0_18_	755.1733	609.1363; 285.0346; 284.0283; 255.0262; 227.0300; 178.9987; 150.9968	Leaves-MeOH Leaves-infusion	[48]
46.	Di-*O*-caffeoylcatalpol isomer 1	28.766	C_33_H_34_O_16_	685.1722	523.1403; 323.0718; 179.0304; 161.0213	Leaves-MeOH	[33]
47.	Kaempferol-*O*-(pentoside-hexoside)-*O*-deoxyhexoside	29.306	C_32_H_38_O_19_	725.1650	579.1321; 284.0289; 145.0291	Leaves-MeOH Leaves-infusion	[49]
48.	Di-*O*-caffeoylcatalpol isomer 2	29.997	C_33_H_34_O_16_	685.1712	523.1397; 343.0764; 179.0345; 161.0159	Leaves-MeOH	[33]
49.	6-*O*-(*E*)-caffeoylajugol	30.552	C_24_H_30_O_12_	509.1973	347.1728; 179.0338; 161.0230	Leaves-MeOH Leaves-infusion	[31]
50.	Quercetin	31.148	C_15_H_10_O_7_	301.0324	178.9964; 151.0009; 121.0308; 107.0146	Leaves-MeOH	[43]
51.	Luteolin	31.176	C_15_H_10_O_6_	285.0428	267.0399; 241.0546; 175.0375; 151.0058; 133.0311	Leaves-MeOH Leaves-infusion	[36,38]
52.	Trihydroxy-octadecadienoic acid	32.36	C_18_H_32_O_5_	327.2116	291.1989; 229.1460; 211.1336; 171.1031	Leaves-MeOH Leaves-infusion Stem bark-MeOH Stem bark-infusion	[45]
53.	Trihydroxy-octadecenoic acid	33.91	C_18_H_34_O_5_	329.2288	311.2203; 293.1239; 229.1450; 211.1334; 171.1011;	Leaves-MeOH Leaves-infusion Stem bark-MeOH Stem bark-infusion	[45]
54.	Apigenin	34.088	C_15_H_10_O_5_	269.0417	227.0342; 151.0027; 117.0349; 107.0126	Leaves-MeOH Leaves-infusion	[40,43]
55.	Tetrahydroxyflavone	34.652	C_15_H_10_O_6_	285.0365	151.0018; 133.0281; 117.0336	Leaves-MeOH Leaves-infusion	[44]
56.	Spathodic acid	43.761	C_30_H_48_O_5_	487.3429	469.2759; 443.2390	Leaves-MeOH Stem bark-MeOH Stem bark-infusion	[5,41]
57.	Hydroxy octadecatrienoic acid	47.369	C_18_H_30_O_3_	293.2078	275.2015; 224.1403; 195.1388	Leaves-MeOH Leaves-infusion Stem bark-MeOH	[38]

Rt—retention time.

### 2.2. Antioxidant Capacity

Over the last century, many studies have shown that there is an inverse relationship between antioxidant intake and various health problems, such as cancer, cardiovascular disease, and diabetes [50,51,52,53]. Antioxidants can be described as powerful shields against the negative effects of free radicals. In this sense, the discovery of new, safe, and effective antioxidants could provide us with new weapons in this war [54]. Thus, plants are one of the most important antioxidant treasures. In view of the above points, the antioxidant capacities of *S. campanulata* extracts were tested via various in vitro assays, including free radical scavenging (DPPH and ABTS), reducing power (CUPRAC and FRAP), metal chelation, and phosphomolybdenum. The results are shown in Table 1. Two radical scavenging assays, ABTS and DPPH, were performed to assess radical quenching. In both assays, the leaves-MeOH extract (DPPH: 177.48 mg TE/g; ABTS: 186.22 mg TE/g) was the most potent, followed by the leaves-infusion, stem bark-infusion and stem bark-methanol extracts. The reducing capacity of antioxidant compounds is closely related to their electron-donating abilities. High electron donating abilities indicate strong antioxidant capacities. For this purpose, cupric (CUPRAC) and ferric (FRAP) reducing power assays were performed. The most robust reducing ability was determined in the leaves-MeOH extract (CUPRAC: 329.69 mg TE/g; FRAP: 220.23 mg TE/g), while the weakest reducing ability was found in the stem bark-MeOH extract (CUPRAC: 63.07 mg TE/g; FRAP: 43.23 mg TE/g) in both assays. The phosphomolybdenum assay is a total antioxidant assay based on the reduction of Mo (VI) to Mo (V) by antioxidants under acidic conditions. In this assay, the potency of the tested extracts was ranked as: leaves-MeOH (2.58 mmol TE/g) > leaves-infusion (1.32 mmol TE/g) > stem bark-infusion (1.14 mmol TE/g) > stem bark MeOH (0.97 mmol TE/g). Transition metals have important redox potentials and play central roles in the Fenton and Haber–Weiss reactions, which generate hydroxyl radicals. Thus, the chelation of transition metals can be considered an important pathway in antioxidant mechanisms. As seen in Table 1, the greatest metal chelating ability was observed in the leaves-infusion extract, with 33.95 mg EDTAE/g, followed by the leaves-methanol (25.30 mg EDTAE/g), stem bark-infusion (18.61 mg EDTAE/g), and stem bark-MeOH (3.15 mg EDTAE/g) extracts. When all of the antioxidant capacity tests were evaluated together, the leaf extracts exhibited stronger effects than the stem bark extracts. With the exception of the metal chelating assay, the leaves-MeOH extract showed the greatest activity, and the capabilities were consistent with the total phenolic levels of the extracts. The number of compounds identified in the leaf extracts was higher than that of the stem bark extracts (Figure 2a). In addition, five components (isoquercetin, quercetin-*O*-dihexoside, di-*O*-caffeoylcatalpol isomer 1, di-*O*-caffeoylcatalpol isomer 2, and quercetin) were identified in the leaves-MeOH extract only; the observed abilities of this extract could be attributed to the presence of these components (Figure 2b). Moreover, they have been described as powerful antioxidants in previous studies [55,56,57,58]. In addition, the total phenolic content of the tested extracts was strongly correlated with the antioxidant properties (Figure 2c). The antioxidant abilities of *S. campanulata* extracts have been reported in the literature by several researchers. In a recent study by Santos, Minatel, Lima, Silva, and Chen [13], who investigated the antioxidant effects of leaves, nectar and flowers of *S. campanulata* using DPPH, FRAP, and ORAC assays, the strongest abilities were determined in leaf extracts. Villarreal, Moreno, Jaimez, Rojas-Fermin, Lucena, Diaz, Diaz, and Carmona [14] determined the antioxidant properties of some Bignoniaceae members, including *S. campanulata*, and the flower extract of *S. campanulata* showed the greatest DPPH scavenging ability and the lowest IC_50_ value. In contrast to our results, the leaf extracts of *S. campanulata* in a previous study [12] had lower antioxidant properties than the flower extracts. These contradictory results can be explained by differences in the collection location, soil composition, and climatic conditions.

### 2.3. Enzyme Inhibitory Properties

The human population has nearly doubled since the last century, and—in proportion to this—the prevalence of some diseases is increasing every day. Given this fact, we need effective and safe therapeutics. Enzymes are one of the cornerstones in the development of therapeutics. Enzymes are the targets of most therapeutic approaches, and their inhibition could alleviate the pathological conditions observed in diseases such as Alzheimer′s, obesity, or type 2 diabetes [59]. For example, amylase and glucosidase are key players in controlling blood sugar levels in diabetics, and their inhibition could help to control blood sugar levels after a high-carbohydrate diet [60]. A similar link also exists between lipase and obesity [61,62]. From this standpoint, safe, potent, and inexpensive enzyme inhibitors are gaining great interest for pharmaceutical and medical applications, and plants, particularly plant secondary metabolites, have shown potent enzyme inhibitory activity in recent studies. With this in mind, the enzyme inhibitory properties of *S. campanulata* extracts against amylase, glucosidase, tyrosinase, and cholinesterases (AChE and BChE) were tested. The results are given in Table 1. In the anti-amylase inhibition assays, the tested methanol extracts exhibited stronger effects than the infusions, and the greatest effect was demonstrated by the leaves-MeOH extract, with 0.53 mmol ACAE/g. Similar observations were also recorded for the glucosidase inhibition assay, in which the order was: stem bark-MeOH (3.82 mmol ACAE/g) > leaves-MeOH (2.77 mmol ACAE/g) > stem bark-infusion (2.74 mmol ACAE/g) > leaves-infusion (0.85 mmol ACAE/g). Tyrosinase catalyses the production of melanin, and its inhibition is an important mechanism in controlling hyperpigmentation problems [63]. As can be seen from Table 1, the highest tyrosinase inhibitory effects were detected in the methanol extracts (stem bark: 64.41 mg KAE/g; leaves: 59.72 mg KAE/g). The weakest anti-tyrosinase effects were found in the stem bark infusion extract (5.10 mg KAE/g). For AChE inhibition, the tested methanol extracts (leaves: 1.88 mg GALAE/g; stem bark: 1.85 mg GALAE/g) showed inhibitory effects, while the infusions did not actively inhibit AChE. Interestingly, the tested leaf extracts did not actively inhibit BChE, whereas the stem bark extracts inhibited the enzymes (MeOH: 6.98 mg GALAE/g; infusion: 1.20 mg GALAE/g). Clearly, the enzyme inhibitory effects were not correlated with the total bioactive compounds in the tested extracts (Figure 2c). Some components may have contributed to the observed enzyme inhibitory effects. For example, caffeic acid, rutin, quercetin, verminoside, and spathodic acid can play the role of inhibitors in these assays. Consistent with our approach, most of these compounds have previously been reported as potent inhibitors [64,65,66,67,68]. To the best of our knowledge, there are no scientific results on the enzyme inhibitory effects of *S. campanulata* extracts. Thus, our results are the first in the literature, and they could provide valuable hints for the further application of *S. campanulata*.

### 2.4. Antineoplastic and Antiviral Ability

The *S. campanulata* extracts showed low cytotoxicity towards kidney fibroblasts, with CC_50_ values above the highest tested concentration of 1000 µg/mL (Table 3). Similar results were obtained for the leaf infusion (ScLI) with HeLa and RKO cells and the stem bark infusion (ScSbI) with RKO cells. The *S. campanulata* stem bark methanolic extract (ScSbM) showed the highest cytotoxicity (CC_50_ 119.03 µg/mL) against HeLa cells. Moreover, this extract exerted selective anticancer activity against all tested cancer cell lines (Figure 3A,B)—FaDu, HeLa, and RKO—with SI values of 6.15, 8.4, and 4.5, respectively. When tested on VERO cells, ScSbM was non-toxic up to 500 µg/mL (Figure 3A). Figure 3 C shows a comparison between VERO, RKO, and HeLa cells treated with ScSbM at 250 µg/mL; it can be clearly seen that the VERO cells were not affected, whereas the RKO monolayer showed lower confluency and differences in cellular morphology in comparison to the control cells, indicating a cytotoxic effect. However, the most noticeable toxic impact was observed in the HeLa cells, resulting in the destruction of the cellular monolayer. The *S. campanulata* stem bark infusion (ScSbI) showed selectivity only against FaDu and HeLa cells, whereas the leaf infusion (ScLI) exhibited selectivity exclusively against FaDu cells (Figure 3B).

*Spathodea campanulata* leaves and stem bark extracts have been reported to be used in traditional medicine to treat cancer [69,70]. Decoctions from bark and leaves are traditionally used in Ghana for the treatment of stomach, skin, and throat cancers [29]. *S. campanulata* bark and root extracts showed cytotoxicity towards leukaemia (CCRF-CEM) cells, with CC_50_ values of 63.29 and 58.08, respectively [11]. Hexane concentrates of *S. campanulata* buds and flowers showed anticancer activity against breast (MCF7) and colon (HCT116) cancers, with reported CC_50_ values between 4.2 and 15.6 µg/mL in a sulforhodamine B stain (SRB) assay. The essential oils obtained from buds and flowers exerted lower activity on these cell lines (CC_50_ 20.2–23 µg/mL) [71]. The flower methanolic extract of *S. campanulata* inhibited glioma (U373), lung cancer (A549), and melanoma (SKMEL-28), with CC_50_ values between 76 and 81 µg/mL determined using an MTT colourimetric assay [15]. Unfortunately, no normal cells were used in the research mentioned above, and selectivity was not evaluated.  This is why our study aimed to fill this scientific gap, showing not only that *S. campanulata* inhibits cancer cell viability but also proving the selective anticancer activity of tested samples. In our studies, the *S. campanulata* extracts were not toxic to normal kidney fibroblasts (VERO). In vivo studies described in the literature support this observation, e.g., an *S. campanulata* leaf methanolic extract was reported not to induce acute toxicity when administered orally (500–4000 mg/kg) to mice [19].

In order to assess the antiviral potential of *S. campanulata,* we selected HHV-1 and CVB3, which are dsDNA and ssRNA viruses, respectively. The tested extracts were incubated with the VERO cells infected with the appropriate virus, and their influence on the occurrence of the CPE was evaluated and compared with the virus control (VC; virus-infected but untreated cells). As is shown in Figure 4C–F, the *S. campanulata* extracts did not inhibit the cytopathic effect exerted by CVB3 in the VERO cell line; thus, it can be concluded that the tested extracts do not show activity towards this virus. Subsequent endpoint CVB3 titration screening showed no significant reduction in virus titres in the tested samples when compared with the titre in the VC. However, when tested on HHV-1-infected cells, the *S. campanulata* leaf methanolic extract (ScLM) at a concentration of 500 µg/mL managed to suppress the formation of the cytopathic effect (Figure 5C). Even at a halved concentration of 250 µg/mL, ScLM noticeably inhibited (Figure 5D) the CPE in comparison to the VC (Figure 5B). Interestingly, SCSbM also managed to decrease the development of the CPE at both tested concentrations (500 and 250 µg/mL) (Figure 5E,F). The HHV-1-infected cells treated with ScLI and ScSbI (Figure 5G,H) showed a CPE comparable to the VC (Figure 5B).

To further evaluate the anti-HHV-1 activity of *S. campanulata,* the collected samples were subjected to an endpoint dilution assay to measure the HHV-1 infectious titre. The reduction in the HHV-1 infectious titre (Δlog) in comparison to the VC is presented in Table 4. During all endpoint titrations performed, the infectious titre of HHV-1 in the samples treated with ScLM 500 µg/mL could not be measured (an example of the titration assay is shown in Figure 6); taking into account the mean HHV-1 infectious titre in the VC, the mean inhibition was 5.11 log (Table 4). On the other hand, at a concentration of 250 µg/mL, ScLM reduced the viral infectious titre by 4.22 log. Since a reduction in viral infectious titre by at least 3 log is required for significant antiviral activity, ScLM can be regarded as having such activity. Among the rest of the *S. campanulata* extracts, only ScSbM showed inhibition higher than 1 log; however, this was not high enough to be considered significant. Acyclovir, used as a model antiviral drug, not only inhibited the development of the CPE but also abolished the infectivity of HHV-1 (Δlog > 5).

Since *S. campanulata* showed significant anti-HHV-1 activity by reducing viral infectious titre, the next step was to evaluate how the extract influenced the replication of viral DNA. For this purpose, DNA was isolated from samples collected from the antiviral assays, and real-time PCR was used to amplify (Figure 7A) the HHV-1-specific sequence of the UL54 gene, which is responsible for transcription regulation. The quantification of viral load was possible owing to a calibration curve (Figure 7B) prepared from standard HHV-1 DNA isolates that were previously analysed using an IVD-certified diagnostic test. The melt curve analysis (Figure 7C) confirmed that a specific amplicon was present in all samples that tested positive during PCR amplification. Interestingly, ScLM at 500 µg/mL was the only sample that showed a noticeable reduction in viral load by 1.45 log (Table 4). The fact that a significant reduction was observed in the HHV-1 viral infectious titre (5.11 log) with ScLM alongside a reduction in viral load of only 1.45 log may indicate that viral replication was mainly inhibited during the stages of the replication cycle involving the production of structural proteins or the maturation of viral progeny, rather than during the replication of viral DNA.

Padhy [29] presented a review of the ethnomedicinal, phytochemical, and pharmacological profile of *Spathodea campanulata* and reported that a preparation from the bark of *S. campanulata* and *Pteleopsis hylodendron* and the ground stalk of *Costus afer* is used in the traditional medicine of Cameroon for bathing patients with chicken pox [29]. It is worth mentioning that the varicella-zoster virus (VZV), responsible for chicken pox, is a herpesvirus, just like HHV-1 used in our research. Moreover, antiviral drugs, such as acyclovir, used against HHV-1 are also effective against VZV. Bathing in a concoction of the bark of *S. campanulata* and ground stalk of *Costus afer* is also traditionally used in Cameroon for the treatment of genital herpes caused by HHV-2, whereas in Western Nigeria, a preparation from *S. campanulata* leaves is used in the treatment of measles, and in Uganda, decoctions from leaves and stem bark are utilised in the treatment of AIDS [29]. Thus, our study validates the traditional use of *S. campanulata* in treating viral diseases. However, further in vivo studies are necessary to confirm the observed activity.

Interestingly, Anani et al., who investigated Togo′s medicinal plants for antiviral and antimicrobial activities, reported that *S. campanulata* do not exert any significant antiviral activity towards HHV-1, SINV (Sindbis virus), or poliovirus type 1 [72]. It is difficult to say why the authors observed no antiviral activity. A possible explanation could be a different origin of the tested plants; however, unfortunately, they [72] did not provide a chemical analysis, and we could not compare it with our results. In addition, in the cited study, different concentrations of the tested samples were used, expressed as equivalents (µg/mL) of dried plant material [72], while in our research, we used different concentrations of dried extract (µg/mL) in cell media. Thus, comparing our results with the work mentioned above is impossible.

The *S. campanulata* leaf methanol extract, which exerted significant anti-HHV-1 activity in the present research, showed the presence of some bioactive molecules that have been previously described as having antiviral potential. For example, the methyl gallate was reported to inhibit HHV-1 in low concentrations (IC_50_ 0.64 µg/mL) [73], as well as the HIV-1 virus and HIV-1 enzymes (reverse transcriptase and integrase) [39]. Another compound, isoquercetin, was shown to inhibit influenza virus replication, exert a synergistic effect when used with amantadine, and suppress the emergence of amantadine- or oseltamivir-resistant viruses in co-treatment with other antivirals [74]. Interestingly, isoquercetin was proven to possess broad-spectrum antiviral activity against herpesviruses (HHV-1, HHV-2, VZV, and HCMV (human cytomegalovirus)), Zika, dengue, and Ebola viruses [75,76,77,78]. There is also a report on the potential application of isoquercetin in the treatment and prevention of severe acute respiratory syndrome coronavirus 2 (SARS-CoV-2) infection [76].

### 2.5. Molecular Docking

The binding energies of the bioactive compounds are tabulated in Table 5. Kaempferol 3-*O*-(2-*O*-β-d-xylopyranosyl)-β-d-galactopyranoside, 6-*O*-trans-caffeoyl-asystasioside E (spatheoside B), and quercetin-3-*O*-(2-*O*-β-d-xylopyranosyl)-β-d-galactopyranoside demonstrated a high binding potential for HSV-1 protease. While these compounds bound less tightly to HSV-1 DNA polymerase, they were too big to fit in the catalytic channel of HSV-1 thymidine kinase, and hence were considered non-binders. On the other hand, caffeic acid bound all three target proteins with different strengths.

An analysis of the protein–ligand interaction revealed that kaempferol 3-*O*-(2-*O*-β-d-xylopyranosyl)-β-d-galactopyranoside spanned the HVS-1 protease active site to form multiple interactions, including five H-bonds via hydroxyl groups, two π–anion interactions, hydrophobic interactions, and several van der Waals interactions all over the tunnel (Figure 8). By comparison, the binding mode of 6-*O*-trans-caffeoyl-asystasioside E (spatheoside B) was rather linear, forming an interaction at the entrance to, along, and deep inside the HVS-1 protease active site. Similarly, these interactions comprised three H-bonds, two π–π interactions, four hydrophobic interactions, and several van der Waals interactions (Figure 9). In the case of quercetin-3-*O*-(2-*O*-β-d-xylopyranosyl)-β-d-galactopyranoside, its binding mode resembled that of kaempferol 3-*O*-(2-*O*-β-d-xylopyranosyl)-β-d-galactopyranoside in terms of filling the catalytic channel of the HVS-1 protease. However, different interactions were formed, including four H-bonds, π–π interactions, a hydrophobic contact, and several van der Waals interactions throughout the tunnel (Figure 10). Together, these interactions are likely to inhibit the viral protease.

## 3. Materials and Methods

### 3.1. Plant Materials

The leaves and stem bark of *S. campanulata* samples were collected during August 2019 from Noumassi (Transua, Gontougo Region) in Côte d’Ivoire. Voucher specimens were deposited at the herbarium of the above-mentioned centre. The leaves and stem bark of the plant samples were dried in shade conditions at room temperature for about one week. Then, the powdering procedure was performed using a mill, and the samples were stored in the dark.

The extracts were prepared using methanol and water in this study. Overnight, the plant material (10 g) was macerated at room temperature with 200 mL of solvent (methanol). Finally, the solvent was evaporated from the mixtures. The plant materials (10 g) were placed in 200 mL of boiled water for 15 min before being filtered. The water extracts were lyophilized and stored at 4 °C until further analysis was required.

### 3.2. Total Phenolic and Flavonoid Content

The total concentrations of phenolics and flavonoids were determined as described in our earlier paper [79]. The contents were evaluated as gallic acid equivalent (GAE)/g dry extract and mg rutin equivalent (RE)/g dry extract, respectively.

### 3.3. Chemical Characterization

The liquid chromatography–mass spectrometry (LC-ESI-QTOF-MS) analysis of the methanolic and aqueous extracts obtained from the leaves and stem bark of *S. campanulata* was carried out on an Agilent 1200 HPLC system (Agilent Technologies, Santa Clara, CA, USA) coupled to an ESI-Q-TOF mass spectrometer (Agilent Technologies). The analytical procedure was reported in our earlier paper [38]. Compounds were tentatively identified based on the obtained fragmentation patterns, which were compared with spectral information available in databases and the scientific literature. Additionally, the quantitative profile was evaluated based on UV spectra detected at 254 and 320 nm.

### 3.4. Antioxidant and Enzyme Inhibitory Assays

In the current work, the antioxidant assays used were cupric ion reduction antioxidant capacity (CUPRAC), ferric ion reduction antioxidant power (FRAP), metal chelating ability (MCA), 1,1-diphenyl-2-picrylhydrazyl (DPPH) and 2,2′-azino-bis(3-ethylbenzothiazoline) 6-sulfonic acid (ABTS), and the phosphomolybdenum (PBD) assay. The details of the assays were explained in our previous paper [80]. The results were determined as the equivalents of Trolox (TE) or EDTA (EDTAE) (in the MCA assay). Regarding the enzyme inhibitory assays, different enzymes (amylase, glucosidase, cholinesterases (AChE and BChE), and tyrosinase) were selected and their inhibition was expressed as standard equivalents (acarbose (ACAE) for amylase and glucosidase; galanthamine (GALAE) for cholinesterases; kojic acid (KAE) for tyrosinase). The experimental details of the enzyme inhibition assays were given in our earlier studies [80,81].

### 3.5. Evaluation of Cytotoxicity and Anticancer Selectivity

A cytotoxicity evaluation was performed against normal VERO (ECACC, Cat. No. 84113001; green monkey kidney) and cancer cell lines (FaDu (ATCC, HTB-43, human hypopharyngeal squamous cell carcinoma), HeLa (ECACC, Cat. No. 93021013, cervical adenocarcinoma), and RKO (ATCC, Cat. No. CRL-2577, colon carcinoma)) using a microculture tetrazolium assay (MTT) according to the previously described methodology [46]. In short, the cells were treated in 96-well plates with serial dilutions of *S. campanulata* extracts for 72h, followed by an assessment of cellular viability using the MTT protocol. The details are described in the Appendix A. The data were exported to GraphPad Prism to evaluate the CC_50_ values (50% cytotoxic concentration). Moreover, the selectivity indexes (SI) were assessed in relation to VERO cells (SI = CC_50_VERO/CC_50_Cancer; SI > 1 is an indication of anticancer selectivity).

### 3.6. Evaluation of Antiviral Effects

Antiviral activity towards HHV-1 and CVB3 was tested according to a previously published methodology [46]; details can be found in the Appendix A. *S. campanulata* extracts in concentrations non-toxic to VERO cells were tested for their effect on HHV-1 and CVB3 replication in virus-infected VERO cells. In short, VERO cells growing in 48-well plates were treated with HHV-1 or CVB3 (100-fold CCID_50_; CCID_50_—50% cell culture infectious dose) for 1 h to allow for successful virus attachment and penetration. Subsequently, the cells were washed with PBS (phosphate-buffered saline) and incubated with *S. campanulata* extracts until a cytopathic effect (CPE) was observed in the virus control (VC). Subsequently, the plates were thrice frozen (−72 °C) and thawed, and samples were collected for the virus titration assay and viral load measurements. The virus infectious titre was measured using an endpoint titration assay, whereas viral load was determined using qPCR. In the endpoint dilution assay for HHV-1, the VERO cells in 96-well plates were incubated with tenfold dilutions of samples in cell media for 72 h with a subsequent MTT-based viability assessment. Afterwards, the reduction in HHV-1 infectious titre (Δlog) was calculated (Δlog = logCCID_50_VC − logCCID_50_SE; VC—virus control; SE—Spathodea extract) for every titration assay, and the results were determined as the mean Δlog. A reduction in infectious titre by ≥3 log was considered to indicate significant antiviral activity. The viral load was measured using real-time PCR. HHV-1 DNA was isolated using a QIAamp DNA Kit (QIAGEN GmbH, Hilden, Germany), following the manufacturer’s instructions. The real-time PCR amplification was performed using HOT FIREPol EvaGreen qPCR Mix (Solis BioDyne, Tartu, Estonia) and primers (UL54F-5′ CGCCAAGAAAATTTCATCGAG 3′, UL54R-5′ ACATCTTGCACCACGCCAG 3′) in a CFX96 thermal cycler (Bio-Rad Laboratories, Inc., Hercules, CA, USA). The details of the amplification parameters can be found in the Appendix A. The quantitative analysis was performed with reference to a calibration curve composed of tenfold dilutions of HSV-1 DNA isolate, which were previously analysed using an IVD certified GeneProof Herpes Simplex Virus (HSV-1/2) PCR Kit (Cat#HSV/ISEX/025, GeneProof a.s., Brno, Czech Republic).

### 3.7. Molecular Docking

The crystal structures of herpes simplex virus type 1 (HSV-1) DNA polymerase (PDB ID: 2GV9) [82], HSV-II protease (PDB ID:2GV9) [83], and HSV-1 thymidine kinase (PDB ID: 1E2J) [82] were retrieved from the Protein Data Bank (https://www.rcsb.org/). The proteins were prepared at a physiological pH of 7.4 using the Biovia Discovery Studio (DS) (Accelrys Software Inc., San Diego, CA, USA, 2012), during which atom bond orders were corrected, missing side-chain atoms and hydrogens were added, and energy minimization was carried out. The three-dimensional (3D) structures of caffeic acid, kaempferol 3-*O*-(2-*O*-β-d-xylopyranosyl)-β-d-galactopyranoside, 6-*O*-trans-caffeoyl-asystasioside E (spatheoside B), and quercetin-3-*O*-(2-*O*-β-d-xylopyranosyl)-β-d-galactopyranoside were downloaded from the PubChem database (pubchem.ncbi.nlm.nih.gov/). Their geometries were optimized using the “lig prep” toolkit in the Biovia DS. Docking grid and parameter files were generated using the respective coordinates of each cocrystal ligand in its respective crystal structure using AutodockTools (https://autodock.scripts.edu); Autodock 4.2′s Lamarckian genetic algorithm [84] was used to generate distinct ligand poses in the binding site of each protein. Protein–ligand interactions were visualized using the Biovia DS Visualizer.

### 3.8. Statistical Analysis

This study used ANOVA (Tukey’s test) to determine whether there were any differences in extract levels between the three samples. The statistical analysis was conducted using Xlstat version 2016. Pearson correlation values determined using the R 3.5.1. software were reported to assess the relationship between total bioactive compounds and biological activities.

## 4. Conclusions

The current work reflected the significant biological effects of *S. campanulata* extracts. The extracts contained a wide range of bioactive compounds, particularly iridoids, flavonoids, and phenolic acids (containing a caffeic acid moiety). In general, the tested leaf extracts showed greater radical scavenging and reducing abilities. Regarding solvents, methanol was more active than water in the antioxidant and enzyme inhibition assays. In antineoplastic assays, the leaf infusion and extracts (both methanol and infusion) obtained from stem bark showed significant and selective antineoplastic activities. Moreover, the leaf methanolic extract exerted substantial antiviral activity against HHV-1, inhibiting the development of the CPE and reducing the viral infectious titre by 5.11 log and the viral load by 1.45 log. In general, the observed promising biological effects of the tested extracts were closely related to their chemical profiles. The reported results suggest that the promising biological capabilities of *S. campanulata* extracts could lead to the creation of innovative visions for the development of potent applications as a source of natural bioactive compounds. However, further studies, such as in vivo animal and bioavailability studies, are needed to understand the efficacy and safety of the tested extracts.

## Figures and Tables

**Figure 1 molecules-27-04694-f001:**
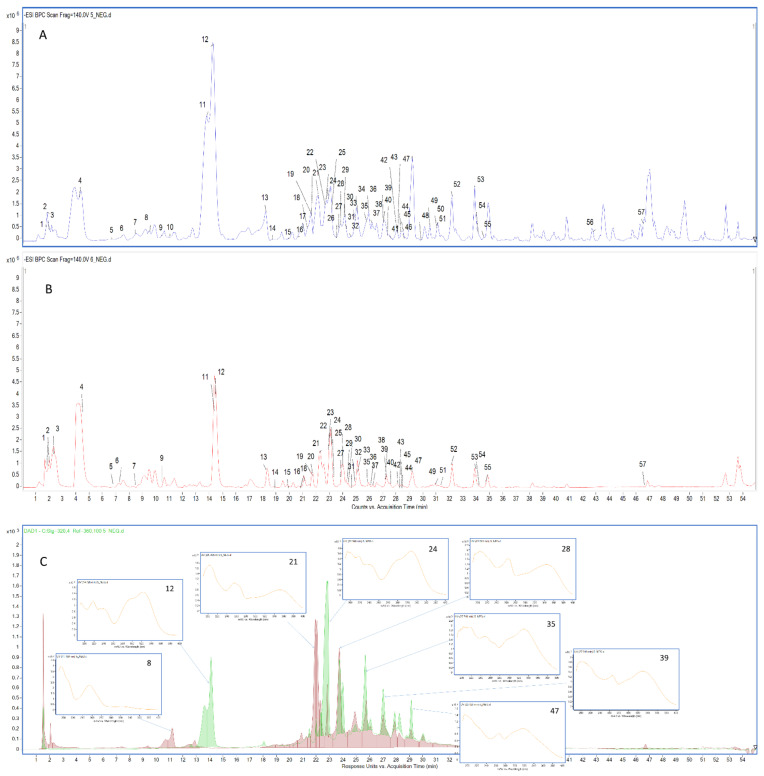
Spectrochromatographic profile of leaf–MeOH (**A**) and leaf infusion (**B**) extracts; numbers correspond to compounds listed in Table 1. (**C**)—UV chromatograms of leaf–MeOH extract: violet—254 nm; green—320 nm. UV spectra for most the abundant compounds are presented.

**Figure 2 molecules-27-04694-f002:**
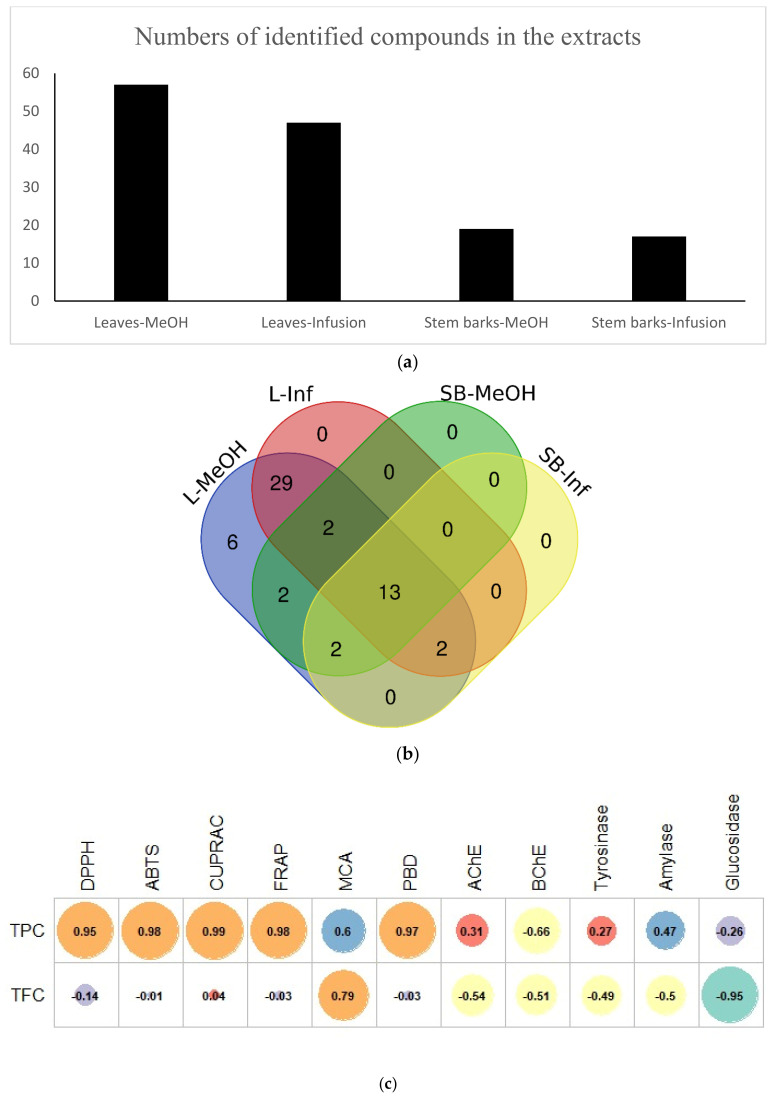
(**a**) Number of identified compounds in the extracts. (**b**) Venn diagram showing the number of common compounds found in the tested extracts. L-MeOH—leaves-MeOH; L-Inf—leaves-infusion; SB-MeOH—stem bark-MeOH; SB-Inf—stem bark-infusion. (**c**) Pearson’s correlation between total bioactive components and antioxidant and enzyme inhibitory effects.

**Figure 3 molecules-27-04694-f003:**
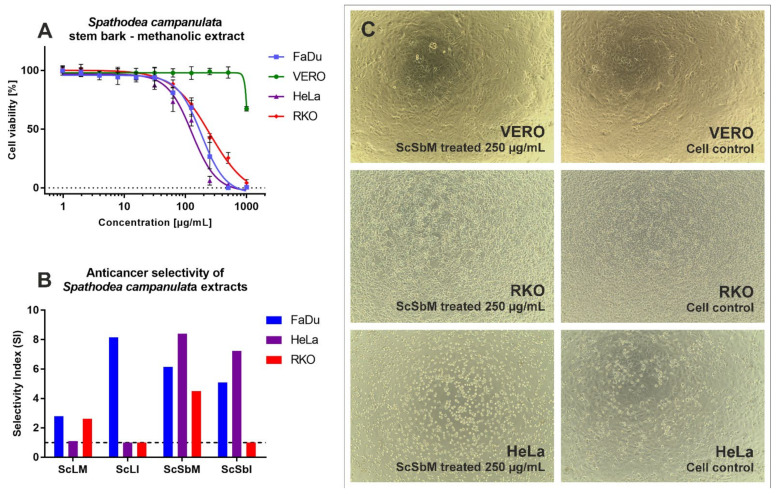
The influence of *S. campanulata* extracts on normal and cancer cells: (**A**)—dose–response curves of ScSbM in normal and cancer cells; (**B**)—comparison of selectivity indexes obtained for *S. campanulata* extracts; (**C**)—influence of ScSbM 250 µg/mL on VERO, RKO, and HeLa cells (magnification × 100).

**Figure 4 molecules-27-04694-f004:**
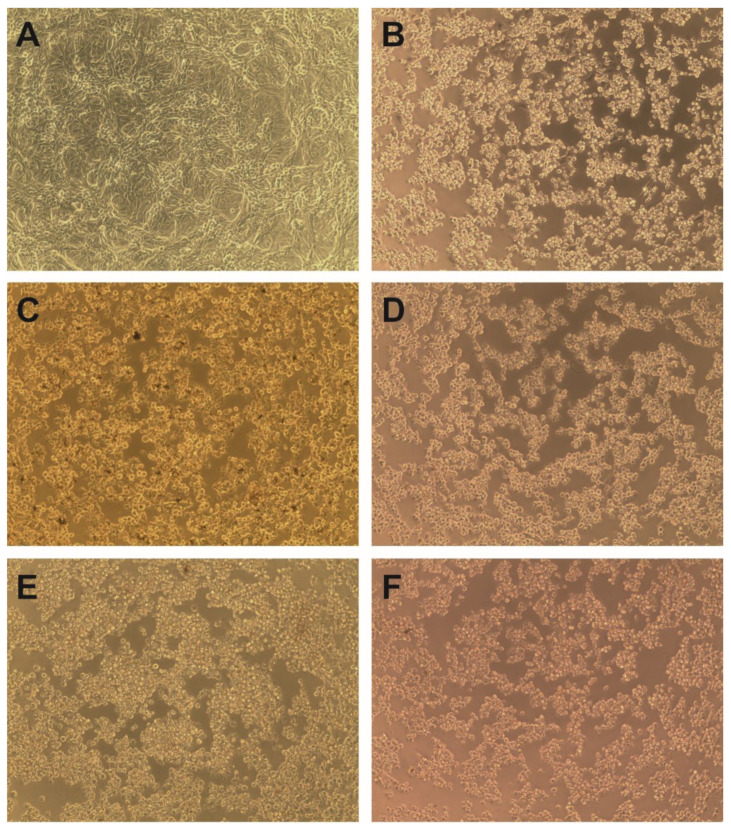
Influence of *S. campanulata* on the CVB3-induced cytopathic effect (magnification × 100). (**A**)—VERO cell control; (**B**)—CVB3 induced cytopathic effect (CPE) in infected VERO cells; CVB3 infected VERO cells treated with ScLM 500 µg/mL (**C**), ScLI 125 µg/mL (**D**), ScSbM 500 µg/mL (**E**) or ScSbI 125 µg/mL (**F**).

**Figure 5 molecules-27-04694-f005:**
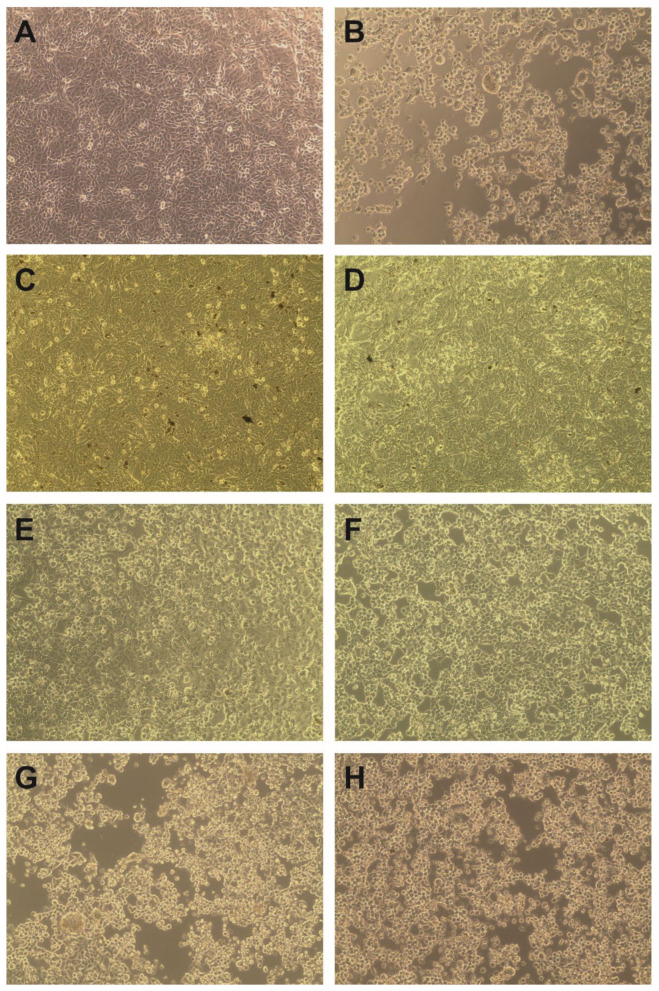
Influence of *S. campanulata* on the HHV-1-induced cytopathic effect (magnification × 100). (**A**)—VERO cell control; (**B**)—HHV-1 induced cytopathic effect (CPE) in infected VERO cells; HHV-1 infected VERO cells treated with ScLM 500 µg/mL (**C**), ScLM 250 µg/mL (**D**), ScSbM 500 µg/mL (**E**), ScSbM 250 µg/mL (**F**), ScLI 125 µg/mL (**G**), or ScSbI 125 µg/mL (**H**).

**Figure 6 molecules-27-04694-f006:**
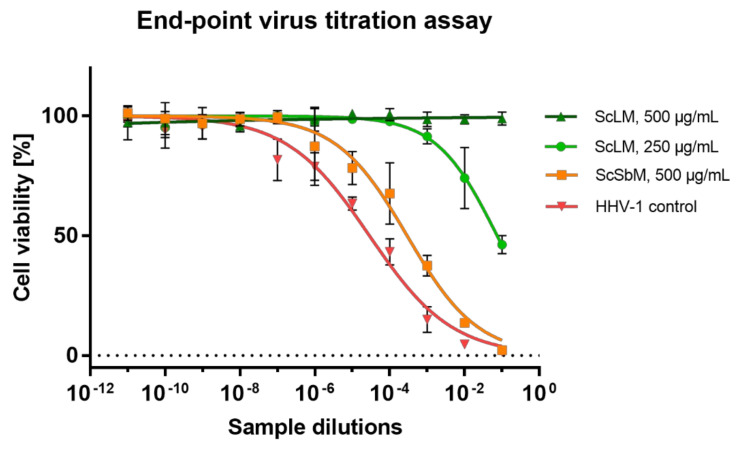
Endpoint dilution assay of HHV-1 infectious titre in virus-infected VERO cells treated with *S. campanulata* extracts.

**Figure 7 molecules-27-04694-f007:**
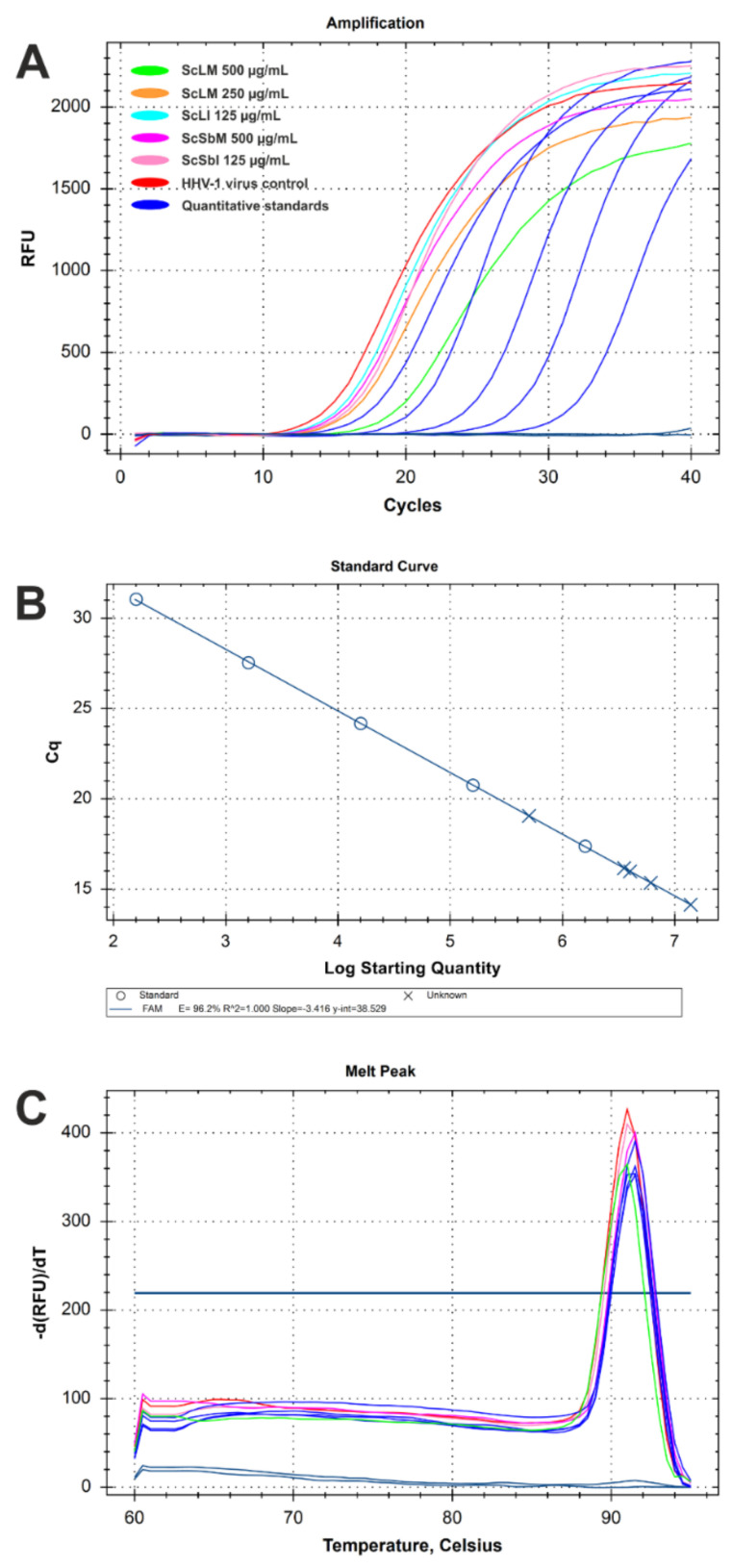
Real-time PCR analysis of HHV-1 viral load in virus-infected VERO cells treated with *S. campanulata* extracts: (**A**)—real-time PCR amplification curves; (**B**)—calibration curve for quantitative analysis; (**C**)—post-amplification melt curve analysis.

**Figure 8 molecules-27-04694-f008:**
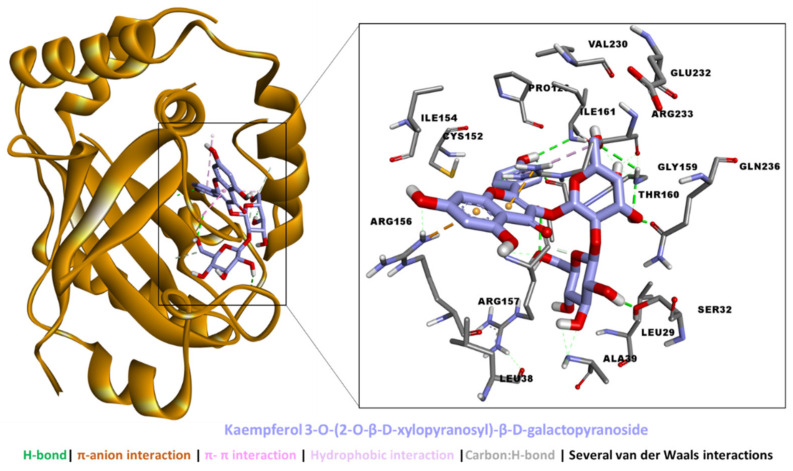
Interaction between HSV-1 protease and kaempferol 3-*O*-(2-*O*-β-d-xylopyranosyl)-β-d-galactopyranoside.

**Figure 9 molecules-27-04694-f009:**
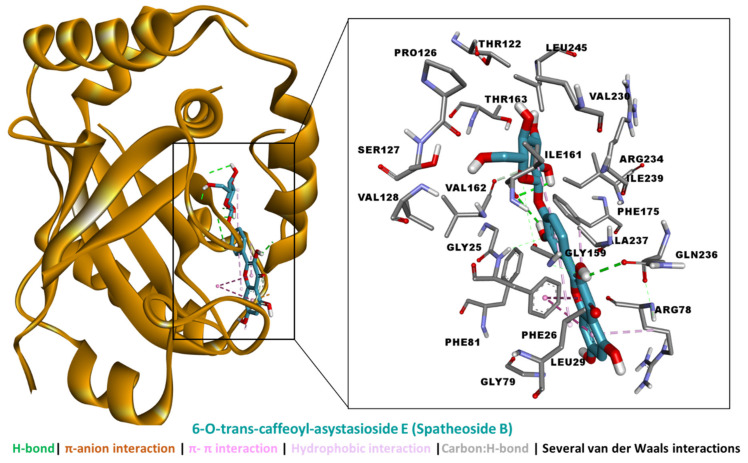
Interaction between HSV-1 protease and 6-*O*-trans-caffeoyl-asystasioside E (spatheoside B).

**Figure 10 molecules-27-04694-f010:**
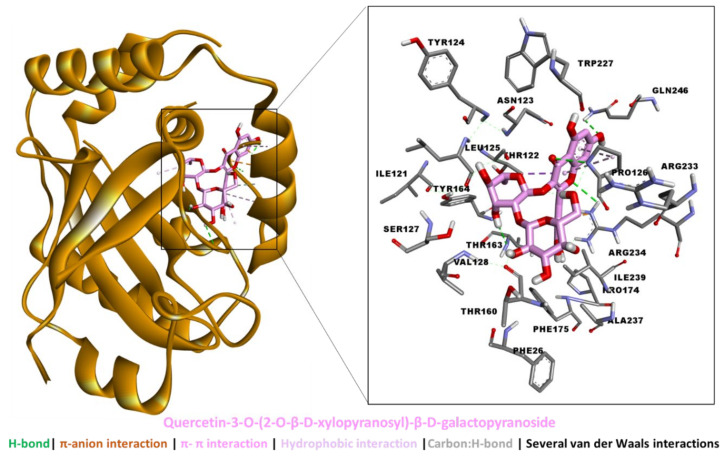
Interaction between HSV-1 protease and quercetin-3-*O*-(2-*O*-β-d-xylopyranosyl)-β-d-galactopyranoside.

**Table 1 molecules-27-04694-t001:** Extraction yields, total bioactive components, and antioxidant and enzyme inhibitory effects of the tested extracts.

Assays	Leaves-MeOH	Leaves-Infusion	Stem Bark-MeOH	Stem Bark-Infusion
** *Extraction Yields (%)* ** ** *Total bioactive compounds* **	9.03	7.15	4.65	4.44
Total phenolic content (mg GAE/g)	89.39 ± 1.69 ^a^	50.31 ± 0.38 ^a^	23.58 ± 0.64 ^d^	31.72 ± 0.41 ^c^
Total flavonoid content (mg RE/g)	6.32 ± 0.01 ^b^	26.96 ± 0.09 ^a^	2.38 ± 0.15 ^d^	3.70 ± 0.21 ^c^
** *Antioxidant assays* **				
DPPH radical scavenging (mg TE/g)	177.48 ± 1.77 ^a^	42.90 ± 0.03 ^b^	31.90 ± 0.19 ^c^	33.37 ± 1.81 ^c^
ABTS radical scavenging (mg TE/g)	186.22 ± 3.52 ^a^	80.02 ± 1.11 ^b^	49.33 ± 0.48 ^d^	60.24 ± 0.89 ^c^
CUPRAC (mg TE/g)	329.69 ± 7.10 ^a^	131.72 ± 1.78 ^b^	63.07 ± 2.66 ^d^	76.04 ± 1.41 ^c^
FRAP (mg TE/g)	220.23 ± 2.50 ^a^	76.91 ± 1.70 ^b^	43.23 ± 1.01 ^d^	47.97 ± 0.95 ^c^
Metal chelating (mg EDTAE/g)	25.30 ± 0.98 ^b^	33.95 ± 0.28 ^a^	3.15 ± 0.30 ^d^	18.61 ± 0.48 ^c^
Phosphomolybdenum (mmol TE/g)	2.58 ± 0.15 ^a^	1.32 ± 0.04 ^b^	0.97 ± 0.10 ^c^	1.14 ± 0.03 ^bc^
** *Enzyme inhibitory assays* **				
AChE inhibition (mg GALAE/g)	1.88 ± 0.22	na	1.85 ± 0.19	na
BChE inhibition (mg GALAE/g)	na	na	6.98 ± 1.04 ^a^	1.20 ± 0.06 ^b^
Tyrosinase inhibition (mg KAE/g)	59.72 ± 1.53 ^b^	10.94 ± 0.97 ^c^	64.41 ± 0.22 ^a^	5.10 ± 0.66 ^d^
Amylase inhibition (mmol ACAE/g)	0.53 ± 0.01 ^a^	0.13 ± 0.01 ^c^	0.43 ± 0.03 ^b^	0.12 ± 0.01 ^c^
Glucosidase inhibition (mmol ACAE/g)	2.77 ± 0.02 ^b^	0.85 ± 0.03 ^c^	3.82 ± 0.07 ^a^	2.74 ± 0.17 ^b^

Values are reported as the mean ± S.D of three parallel measurements. GAE—gallic acid equivalent; RE—rutin equivalent; TE—Trolox equivalent; EDTAE—EDTA equivalent; GALAE—galantamine equivalent; KAE: kojic acid equivalent; ACAE—acarbose equivalent; na—not active. Different letters in the same line indicate significant differences in the extracts (*p* < 0.05).

**Table 3 molecules-27-04694-t003:** Results of cytotoxicity evaluation.

*Spathodea campanulata*	Solvent	Sample	CC_50_ ± SD (µg/mL)			
			VERO	FaDu	HeLa	RKO
Leaves	MeOH	ScLM	>1000	358 ± 11.88	914.07 ± 69.81	382.07 ± 35.86
	Water	ScLI	>1000	122.65 ± 19.45	>1000	>1000
Stem bark	MeOH	ScSbM	>1000	162.55 ± 15.49	119.03 ± 20.72	222.07 ± 17.27
	Water	ScSbI	>1000	196.5 ± 25.74	137.97 ± 18.41	>1000

**Table 4 molecules-27-04694-t004:** Influence of *S. campanulata* extracts on HHV-1 infectious titre and viral load in virus-infected VERO cells.

*S. campanulata*	Solvent	Sample	Concentration (µg/mL)	Reduction in HHV-1 Infectious Titre (Δlog) *	Reduction in HSV-1 Viral Load (Δlog’) **
Leaves	MeOH	ScLM	500	5.11 ± 0.93	1.45 ± 0.13
			250	4.22 ± 1.06	0.38 ± 0.29
	Water	ScLI	125	0.16 ± 0.21	0.31 ± 0.03
			62.5	0.08 ± 0.07	0.28 ± 0.05
Stem bark	MeOH	ScSbM	500	1.19 ± 0.46	0.28 ± 0.1
			250	0.41 ± 0.33	0.16 ± 0.15
	Water	ScSbI	125	0.71 ± 0.19	0.17 ± 0.24
			62.5	0.28 ± 0.06	0.18 ± 0.1

* Δlog (mean ± SD)—calculated from separate titration assays, Δlog = logCCID_50_VC − logCCID_50_FE; VC—virus control; SE—Spathodea extract, Δlog ≥ 3 is regarded as significant; ** Δlog’ (mean ± SD)—calculated for samples originating from separate antiviral assays, Δlog’ = log(copies/mL)VC − log(copies/mL)SE; VC—virus control; SE—Spathodea extract.

**Table 5 molecules-27-04694-t005:** The binding energy of bioactive compounds for the HSV-1 target proteins. No possible binding poses of kaempferol 3-*O*-(2-*O*-β-d-xylopyranosyl)-β-d-galactopyranoside, 6-*O*-trans-caffeoyl-asystasioside E (spatheoside B), or quercetin-3-*O*-(2-*O*-β-d-xylopyranosyl)-β-d-galactopyranoside to HSV-II thymidine kinase were found.

	Binding Energy (Kcal/mol)		
Compounds	HSV-1 DNA Polymerase	HSV-1 Protease	HSV-1 Thymidine Kinase
Caffeic acid	−3.39	−5.60	−8.21
Kaempferol 3-*O*-(2-*O*-β-d-xylopyranosyl)-β-d-galactopyranoside	−7.37	−10.10	-
6-*O*-trans-caffeoyl-asystasioside E (Spatheoside B)	−6.86	−10.77	-
Quercetin-3-*O*-(2-*O*-β-d-xylopyranosyl)-β-d-galactopyranoside	−7.20	−10.44	-

## Data Availability

Not applicable.

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
