# Peer review of "Bridging the Chemical Profiles and Biological Effects of Spathodea campanulata Extracts: A New Contribution on the Road from Natural Treasure to Pharmacy Shelves"

_molecules, 2022, doi:10.3390/molecules27154694_

Round 1

Reviewer 1 Report

The manuscript by Świątek et al. is really interesting and well written. The results are clear and extensively discussed. Minor revisions are necessary for publication of the research article in the journal.

1. Subheadings should be correctly numbered throughout the text.

2. Information on the yield of extractions should be added at the beginning of the Results section.

3. Figure 3C: The images shown are of poor resolution; they should be replaced with higher resolution images. In addition, the influence of ScSbM on FaDu cells should also be shown.

4. Paragraph "Evaluation of cytotoxicity and anticancer selectivity": cell lines should be written in full; also it should be indicated where they were purchased or who supplied them.

Author Response

  1. Subheadings should be correctly numbered throughout the text.

Dear Reviewer, thank you for your watchfulness. Subheadings have been corrected throughout the text.

  1. Information on the yield of extractions should be added at the beginning of the Results section.

Dear Reviewer, we are sorry for not providing the extraction yields. Appropriate corrections have been made in Table 1 of the revised manuscript.

  1. Figure 3C: The images shown are of poor resolution; they should be replaced with higher resolution images. In addition, the influence of ScSbM on FaDu cells should also be shown.

Dear Reviewer, thank you for your valuable comment. Figure 3 has been replaced with a higher resolution copy. Concerning the influence of ScSbM on FaDu cells, unfortunately, we have collected photographic documentation only for VERO, RKO, and HeLa cells. Cell lines are being routinely observed using an inverted microscope in our laboratory, however, it is not possible to document all observations. We are sorry for not being able to comply with your request.

  1. Paragraph "Evaluation of cytotoxicity and anticancer selectivity": cell lines should be written in full; also it should be indicated where they were purchased or who supplied them.

Dear Reviewer, thank you for your remarks and efforts in improving our manuscript. All our cell lines and viruses originate from ATCC or ECACC. Cell lines’ full names and origins have been added to paragraph “3.5. Evaluation of cytotoxicity and anticancer selectivity”.

Reviewer 2 Report

The present work reports the chemical profiles and biological effects of the extracts (methanolic and water) from leaves and stem barks of S. campanulata. As described, tested extracts were chemically dereplicated using LC-ESI-QTOF-MS. Biological effects were tested regarding antioxidants (radical scavenging, reducing power, and metal chelating), enzyme inhibitory (cholinesterase, amylase, glucosidase, and tyrosinase), antineoplastic and antiviral activities. This work was well conducted and reported interesting results. However, before its final acceptance in Molecules, I suggest that the HPLC profile of aqueous (infusion) extract should be presented comparing the obtained profile with that shown in the manuscript for MeOH extract - including also a more detailed discussion concerning their chemical compositions in the text. Additionally, the authors must explain why it was not possible a complete description of the chemical composition of stem bark extracts, as indicated in figure 2a.

Author Response

Reviewer 2: I suggest that the HPLC profile of aqueous (infusion) extract should be presented comparing the obtained profile with that shown in the manuscript for MeOH extract - including also a more detailed discussion concerning their chemical compositions in the text. Additionally, the authors must explain why it was not possible a complete description of the chemical composition of stem bark extracts, as indicated in figure 2a.

Dear Reviewer, thank you for your suggestion. We added a chromatogram of infusion with the indication of identified compounds. We also provided the additional comment related to differences between methanolic and aqueous extract, as well as an explanation of the poorer content of phenolic compounds in stem bark extracts (as indicated in Figure 2a).